# Association between eczema and major cardiovascular outcomes in population-based studies: a systematic review protocol

Anna Ascott,[1] Ashley M Yu,[2] Morten Schmidt,[3] Katrina Abuabara,[4] Liam Smeeth,[1] Sinéad M Langan[1]

[1]Faculty of Epidemiology and Population Health, London School of Hygiene & Tropical Medicine, London, UK
[2]Faculty of Medicine, University of Ottawa, Ottawa, Ontario, Canada
[3]Department of Clinical Epidemiology, Aarhus University Hospital, Aarhus, Denmark
[4]School of Medicine, University of California, San Francisco, California, USA

**Correspondence to**
Anna Ascott;
annaascott@doctors.org.uk

## ABSTRACT

**Introduction** Chronic inflammatory diseases such as eczema (also known as atopic dermatitis) have been inconsistently linked to cardiovascular disease and stroke in both mechanistic and epidemiological studies. There is a need to review the existing epidemiological data examining the association between eczema and major cardiovascular outcomes, including angina, myocardial infarction, coronary revascularisation, heart failure, cardiac arrhythmias, stroke and cardiovascular death, in order to improve our understanding of the comorbidities of eczema. **Methods and analysis** We will systematically review population-based studies, including cohort, case–control and cross-sectional studies, reporting on the association between eczema and cardiovascular outcomes. We will search Medline, Embase and Global Health, from their date of inception to April 2017, using a comprehensive search strategy formulated with the help of a librarian. Two reviewers will independently screen titles and abstracts in duplicate, followed by independent data extraction and quality assessment. We will group studies by the cardiovascular outcome under study and synthesise them narratively. If sufficient numbers of homogeneous studies are returned, we will perform meta-analyses to obtain pooled effect estimates. Preferred Reporting Items for Systematic Review and Meta-Analysis will be used to inform the reporting of this study.
**Trial registration number** CRD42017060359.

## Strengths and limitations of this study

► This systematic review protocol is reported in line with the Preferred Reporting Items for Systematic Review and Meta-Analysis Protocols guidelines.
► A systematic and broad search strategy planned with the help of a librarian attempts to identify all studies that meet the eligibility criteria.
► Two authors will independently assess search results using an online data management program to minimise bias and errors in screening, data extraction and quality assessment of studies.
► Despite an extensive search, it is still possible that relevant papers may have been missed.
► Studies may not systematically report all outcomes.

## INTRODUCTION

Eczema is an inflammatory skin disease, traditionally considered a disease of childhood. However, eczema may affect around 10% of adults[1 2] and the global prevalence of eczema is increasing.[3] Concurrently, cardiovascular disease is fast becoming a significant cause of mortality and morbidity in both high-income and low-income and middle-income countries. Chronic inflammatory conditions have been linked to cardiovascular disease, in diseases of varying aetiology, from psoriasis and rheumatoid arthritis to HIV.[4–7]

There are a number of lines of evidence supporting an association between eczema and cardiovascular disease. Mechanistic studies suggest that platelet dysfunction and decreased fibrinolysis may contribute to increased clotting in eczema.[8 9] Severe eczema has been associated with increased incidence of coronary artery disease using cardiac CT angiography.[10] In addition, treatments used for eczema may increase cardiovascular risk[11 12]; however, a likely limitation of the studies to be included in this review will be the likely strong collinearity between eczema severity and level of treatment. Factors such as confounding by severity could contribute to an observed increase in cardiovascular disease and stroke in eczema.

Epidemiological studies have inconsistently linked eczema to cardiovascular risk factors, disease and stroke across different populations,[13–16] while previous systematic reviews have found an association between eczema and risk factors for cardiovascular disease, including raised body mass index[17] and childhood type 1 diabetes.[18] There is a need to review the existing epidemiological data linking eczema to major cardiovascular outcomes in order to increase our

understanding of the comorbidities of eczema, and to inform management and prevention strategies at an individual and population level.

The primary objective of this systematic review is to ascertain the association between eczema and major cardiovascular outcomes, including angina, myocardial infarction, coronary revascularisation, heart failure, cardiac arrhythmias, stroke and cardiovascular death, in population-based studies.

The secondary objectives of this review are to answer the following questions:

► Does the strength of the association between eczema and major cardiovascular outcomes increase as severity of eczema increases?
► Does treatment of eczema alter the risk of cardiovascular outcomes (although it may not be possible to completely disentangle the effects of disease severity and treatment as those with severe disease are often defined by their treatment with systemic therapy)?
► Is the association between eczema and major cardiovascular outcomes consistent across different countries?
► Does the association vary for different cardiovascular disease endpoints?
► Is there any modification of effect by age and gender?

## METHODS AND ANALYSIS
### Eligibility criteria
Peer-reviewed, published studies in any language, from any year, are eligible to be included. Studies must be population-based, with an average age of participants >18. Studies may be cohort, case–control or cross-sectional designs, from any healthcare setting, with any length of follow-up. The exposure of interest is eczema (atopic dermatitis). The comparator will be people or person years without eczema. Outcomes are major cardiovascular outcomes, which will include angina, myocardial infarction, coronary revascularisation, heart failure, cardiac arrhythmias, stroke and cardiovascular death.

We will exclude case series (including retrospective clinic populations), ecological studies, reviews and studies in paediatric populations only. Studies of localised or other types of eczema such a hand eczema, and seborrheic or contact dermatitis, are not eligible to be included.

### Literature search
We will search the following electronic databases from their date of inception to April 2017: Medline via Ovid, Embase via Ovid and Global Health. The search strategies were created by a researcher with medical and systematic review training, in conjunction with a librarian with expertise in searching the literature, and reviewed by all authors. Exclusion filters and limits will not be used in the search due to the risk that eligible studies may be inadvertently excluded. The Ovid Medline search strategy is available to view (see online supplementary appendix 1). We will also review the bibliographies of included studies, and contact experts in the field for relevant references.

### Selection of studies and data extraction
Covidence, an online literature reviewing data management program, will be used to facilitate collaboration and data extraction between reviewers. All titles and abstracts resulting from the literature search will be uploaded to Covidence. Duplicates will be removed by AA. Two reviewers (AA and AMY) will independently screen titles and abstracts in duplicate. Full-text articles will be retrieved where studies fulfil inclusion criteria, or where there is any ambiguity of the study's eligibility. Disagreement will be resolved through discussion with a third reviewer where necessary. Additional information will be requested from authors if needed. Review authors will not be blinded to the journal titles or study authors. A PRISMA (Preferred Reporting Items for Systematic Review and Meta-Analysis) flow diagram[19] will document the process of literature selection and reasons for exclusion.

Each reviewer will extract data independently and in duplicate in order to minimise bias and errors, using a standardised data extraction tool, which will be piloted on the first three eligible texts to ensure its suitability. Data will be sought for the following domains:
1. study details: author information, publication year, design, sponsorship, geographical location, healthcare setting, length of follow-up time if relevant, sampling and recruitment methods, period of study, aims and objectives
2. population characteristics: for example, mean and median age, inclusion and exclusion criteria
3. exposure: definition of eczema as an exposure, number of exposed subjects, details of eczema severity and treatment, age at onset of eczema
4. comparators: definition of unexposed subjects, number of comparators
5. outcomes: definition and identification of cardiovascular outcomes (angina, myocardial infarction, coronary revascularisation, heart failure, cardiac arrhythmias, stroke and cardiovascular death), number of subjects with the outcome.

The definition of eczema that the authors state in the original study will be recorded in detail and given due consideration in the assessment of study quality. We predict that the definition of eczema severity will be heterogeneous between studies. For example, severe eczema may be defined by the need for systemic treatment; however, this may be a source of misclassification bias. Therapeutic agents such as systemic corticosteroids may be used for alternative clinical indications, and treatment is an imperfect proxy for defining severity. This uncertainty will be dealt with by including a description of how authors reported their eczema severity definitions in the data extraction stage, by paying particular attention to definitions of severity in the appraisal of quality and in the discussion of the review. Where available, unadjusted and fully adjusted effect estimates will be

recorded, along with details of confounders. Discrepancies will be resolved by discussion, with a third reviewer if necessary.

## Outcomes

The primary outcome is the association between eczema and cardiovascular outcomes, including angina, myocardial infarction, coronary revascularisation, heart failure, cardiac arrhythmias, stroke (all subtypes) and cardiovascular death. Where possible, effect estimates will include ORs, HRs and incident rate ratios, for cardiovascular outcomes in people with eczema compared with those without. Secondary outcomes include variation in the strength of association between severity of eczema and cardiovascular disease, whether the association varies by country, and is altered by the treatment of eczema, by different cardiovascular disease endpoints, or modified by age and gender.

## Quality assessment

Critical appraisal will be independently recorded by reviewers to allow comparison by study quality. Risk of bias will be assessed by considering relevant domains to observational studies, including participant selection, measurement of variables and controlling for confounding, in line with the Cochrane Collaboration's Grading of Recommendations Assessment, Development and Evaluation (GRADE) tool for assessing risk of bias, and the Newcastle-Ottawa Scale, in order to maximise relevance to non-randomised studies. Each domain will be rated with 'high', 'low' or 'unclear' with regard to the risk of bias, with free text explanations. Full results of this quality assessment will be included in the resulting review, with discussion of quality assessment in the narrative data synthesis.

## Data synthesis

We will group studies by the cardiovascular outcome under study and synthesise them narratively. No subgroup analyses are planned. We will only consider information on the interaction between eczema and covariates if this has been formally assessed in the original publication. We will perform meta-analyses on the association between eczema and specific cardiovascular outcomes with the help of a statistician to obtain a pooled effect estimate. We will assess statistical heterogeneity using the $I^2$ statistic. The pooled relative risk and its 95% CI will be calculated using random effects models. If substantial heterogeneity is observed, we will explore the reasons for the heterogeneity in sensitivity analyses. Study characteristics and the effect estimates for the association between eczema and cardiovascular disease will be presented in full, in tabular form. Studies at high risk of bias will not be contained in the synthesis. We will look for publication bias using standard approaches including funnel plots and Egger tests. PRISMA guidelines[19] will be used to report the results of this study.

## ETHICS AND DISSEMINATION

This systematic review protocol was registered with the International Prospective Register of Systematic Reviews (PROSPERO) on 22 April 2016. Any amendments to the protocol will be documented on the PROSPERO site contemporaneously, with full explanation of any change. Ethical approval is not required for this study as it is a systematic review. The results will be submitted for peer-reviewed publication, and for national and international presentation.

**Contributors** AA contributed to the design of the study, was the guarantor, developed the search strategy and the PROSPERO protocol, and drafted the manuscript. AMY contributed to the design of the study, approved the search strategy, and provided critical feedback on the PROSPERO protocol and the final manuscript. MS, KA and LS approved the search strategy, and provided critical feedback on the PROSPERO protocol and the final manuscript. SML contributed to the conception and design of the study, and provided critical feedback on the search strategy, PROSPERO protocol and the final manuscript.

**Funding** Publication of this manuscript was funded by a Wellcome Senior Clinical Fellowship to SML (205039/Z/16/Z). AA was supported by a small grant from the British Association of Dermatologists. LS is funded by a Wellcome Trust Senior Fellowship in Clinical Science. The Wellcome Trust and the British Association of Dermatologists played no role in the development of this study or the protocol.

**Competing interests** AA reports grants from the British Association of Dermatologists during the conduct of the study. LS reports grants from Wellcome Trust during the conduct of the study; grants from Wellcome Trust, Medical Research Council, National Institute for Health Research and the European Union outside the submitted work; personal fees from GSK for advisory work unrelated to the submitted work; grant funding from GSK for academic research unrelated to the submitted work; acts as an unpaid steering committee chair for AstraZeneca for a randomised trial unrelated to the submitted work; and is also a trustee of the British Heart Foundation. SML reports grants from Wellcome Senior Clinical Fellowship in Science (205039/Z/16/Z) during the conduct of the study.

**Provenance and peer review** Not commissioned; externally peer reviewed.

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
