## [Reviewer comments · BMJ Open]

ARTICLE DETAILS

TITLE (PROVISIONAL)	The association between eczema and major cardiovascular outcomes in population based studies: a systematic review protocol
AUTHORS	Ascott, Anna; Yu, Ashley; Schmidt, Morten; Abuabara, Katrina; Smeeth, Liam; Langan, Sinead

VERSION 1 – REVIEW

REVIEWER	Magnus Lindberg Dep of Medical Sciences, School of Health and Medicine, Örebro University, Örebro, Sweden
REVIEW RETURNED	01-Jul-2017

GENERAL COMMENTS	I have two concerns: How will the definition of eczema (Atopic dermatitis) be dealt with? How will the definition of eczema severity be dealt with?
---

REVIEWER	Jonatan Lindh Karolinska Institutet Div. of Clinical Pharmacology Sweden
REVIEW RETURNED	19-Jul-2017

GENERAL COMMENTS	The protocol is clearly presented and seems methodologically sound. The research question addressed is clinically relevant, although rather wide with a large number of cardiovascular outcomes which may have to be dealt with separately in the final review, depending on the amount of heterogeneity encountered. I have only a few minor questions: Page 16, line 33: Secondary outcomes are listed, but it is not entirely clear how these will be addressed. Presumably, subgroup analysis and/or meta-regression could be utilized (using study-level estimates of the covariates), but on page 17 it is stated that subgroup analyses will not be performed. Are you only considering information on the interaction between eczema and covariates (e.g. age and gender) if this has been formally addressed in the original publication? Page 17, line13: "If sufficient numbers of homogenous studies are returned, we will perform a meta-analysis" What criteria will be used to determine that the heterogeneity among studies is low enough and is there a predetermined number of studies required? If possible, will you pool studies with different cardiovascular outcomes (i.e. a composite outcome) or will you only perform separate meta-analyses for each outcome due to the risk of eczema affecting different cardiovascular outcomes differently? Will you make any attempts to detect and quantify publication bias?
---

VERSION 1 – AUTHOR RESPONSE

COMMENTS FROM REVIEWER #1:

Comment 1:

How will the definition of eczema (Atopic dermatitis) be dealt with?

Response 1:

The definition that the authors state in the original study will be recorded in detail, and given due consideration in the assessment of study quality. We will also consider the definitions used when considering combining the results of different studies.

Comment 2:

How will the definition of eczema severity be dealt with?

Response 2:

We predict that the definition of eczema severity will be heterogeneous between studies. In relevant studies, severe eczema may be defined by the need for systemic treatment; however, this may be a source of misclassification bias. Therapeutic agents such as systemic corticosteroids may be used for alternative clinical indications and treatment is an imperfect proxy for severity definition. We have dealt with this uncertainty by including a description of how authors reported that they defined eczema severity in the data extraction stage, and paying particular attention to definition of severity in the appraisal of quality. We will highlight the complexity of defining eczema severity in the discussion of the review.

COMMENTS FROM REVIEWER #2:

Comment 3:

Page 16, line 33: Secondary outcomes are listed, but it is not entirely clear how these will be addressed. Presumably, subgroup analysis and/or meta-regression could be utilized (using study-level estimates of the covariates), but on page 17 it is stated that subgroup analyses will not be performed. Are you only considering information on the interaction between eczema and covariates (e.g. age and gender) if this has been formally addressed in the original publication?

Response 3:

We will only consider information on the interaction between eczema and covariates if this has been formally assessed in the original publication.

Comment 4:

Page 17, line13: "If sufficient numbers of homogenous studies are returned, we will perform a meta-analysis"

What criteria will be used to determine that the heterogeneity among studies is low enough and is there a predetermined number of studies required? If possible, will you pool studies with different cardiovascular outcomes (i.e. a composite outcome) or will you only perform separate meta-analyses for each outcome due to the risk of eczema affecting different cardiovascular outcomes differently? Will you make any attempts to detect and quantify publication bias?

Response 4:

We will assess statistical heterogeneity using the I² statistic. The pooled relative risk and its 95% confidence interval will be calculated using random effects models. I agree that we should not pool outcomes as the mechanisms may differ. I agree that it would be useful to look for publication bias using standard approaches including funnel plots and Egger tests.

Thank you again for these helpful recommendations. We very much look forward to hearing from you.
Yours faithfully,
Anna Ascott on behalf of co-authors

VERSION 2 – REVIEW

REVIEWER	Magnus Lindberg Department of Dermatology, University hospital Örebro, Örebro, Sweden and Department of Medical Sciences, Örebro university, Örebro, weden
REVIEW RETURNED	28-Aug-2017

GENERAL COMMENTS	A nice study
--------------

REVIEWER	Jonatan Lindh Karolinska Institutet Sweden
REVIEW RETURNED	25-Aug-2017

GENERAL COMMENTS	My concerns have been adequately addressed. Thank you.
--